# Healthcare providers' perceived acceptability of a warning signs intervention for rural hospital-to-home transitional care: A cross-sectional study

Mary T. Fox[1,2]*, Jeffrey I. Butler[1,2], Adam M. B. Day[3], Evelyne Durocher[4], Behdin Nowrouzi-Kia[5], Souraya Sidani[1], Ilo-Katryn Maimets[6], Sherry Dahlke[7], Janet Yamada[8]

1 School of Nursing, York University, Toronto, Ontario, Canada, 2 York University Centre for Aging Research and Education, Toronto, Ontario, Canada, 3 Northern Ontario School of Medicine, Thunder Bay, Ontario, Canada, 4 School of Rehabilitation Science, McMaster University, Hamilton, Ontario, Canada, 5 Department of Occupational Science and Occupational Therapy, University of Toronto, Toronto, Ontario, Canada, 6 Steacie Science and Engineering Library, York University, Toronto, Ontario, Canada, 7 Faculty of Nursing, University of Alberta, Edmonton, Alberta, Canada, 8 Daphne Cockwell School of Nursing, Toronto Metropolitan University, Toronto, Ontario, Canada

* maryfox@yorku.ca

**Data Availability Statement:** According to the guidelines of our hospital site's Research Ethics

## Abstract

### Introduction

There is a pressing need for transitional care that prepares rural dwelling medical patients to identify and respond to the signs of worsening health conditions. An evidence-based warning signs intervention has the potential to address this need. While the intervention is predominantly delivered by nurses, other healthcare providers may be required to deliver it in rural communities where human health resources are typically limited. Understanding the perspectives of other healthcare providers likely to be involved in delivering the intervention is a necessary first step to avert consequences of low acceptability, such as poor intervention implementation, uptake, and effectiveness. This study examined and compared nurses' and other healthcare providers' perceived acceptability of an evidence-based warning signs intervention proposed for rural transitional care.

### Methods

A cross-sectional design was used. The convenience sample included 45 nurses and 32 other healthcare providers (e.g., physical and occupational therapists, physicians) who self-identified as delivering transitional care to patients in rural Ontario, Canada. In an online survey, participants were presented with a description of the warning signs intervention and completed established measures of intervention acceptability. The measures captured 10 intervention acceptability attributes (effectiveness, appropriateness, risk, convenience, relevance, applicability, usefulness, frequency of current use, likelihood of future use, and confidence in ability to deliver the intervention). Ratings $\geq 2$ indicated acceptability. Data analysis included descriptive statistics, independent samples t-tests, as well as effect sizes

Board, we are not permitted to post the study data for public consumption. Given that our sample was recruited from small towns and participants' characteristics may be recognizable to others even in de-identified data, we are also unable to share the data. However, researchers interested in using our data may submit a request to the hospital REB to request use of the data. Contact info: Research Ethics Board Health Sciences North Research Institute 56 Walford Road Sudbury, ON, P3E 2H2 Phone: 705-523-7300 Email: reb@hsnsudbury.ca.

**Funding:** This work is supported by the Canadian Institutes of Health Research (funding reference number 163072), awarded to MTF. The Canadian Institutes of Health Research (https://cihr-irsc.gc.ca/e/193.html) had no role in the design of the protocol or in the writing of this manuscript.

**Competing interests:** The authors have declared that no competing interests exist.

to quantify the magnitude of any differences in acceptability ratings between nurses and other healthcare providers.

## Results

Nurses and other healthcare providers rated all intervention attributes > 2, except the attributes of convenience and frequency of current use. Differences between the two groups were found for only three attributes: nurses' ratings were significantly higher than other healthcare providers on perceived applicability, frequency of current use, and the likelihood of future use of the intervention (all p's < .007; effect sizes .58 - .68, respectively).

## Discussion

The results indicate that both participant groups had positive perspectives of the intervention on most of the attributes and suggest that initiatives to enhance the convenience of the intervention's implementation are warranted to support its widespread adoption in rural transitional care. However, the results also suggest that other healthcare providers may be less receptive to the intervention in practice. Future research is needed to explore and mitigate the possible reasons for low ratings on perceived convenience and frequency of current use of the intervention, as well as the between group differences on perceived applicability, frequency of current use, and the likelihood of future use of the intervention.

## Conclusions

The intervention represents a tenable option for rural transitional care in Ontario, Canada, and possibly other jurisdictions emphasizing transitional care.

## Introduction

Hospital-to-home transitional care (TC) is defined as the healthcare services patients receive as they shift from hospital to home and involves interventions aimed at improving post-discharge care management [1,2]. Due to increasing hospital readmissions of older patients with complex health conditions, TC has become a priority in many Western healthcare jurisdictions. In rural communities in Ontario, Canada, the need for effective TC is particularly pronounced because patients have more emergency department visits and hospital readmissions during the first 30 days after discharge than urban patients [3]. Compared to urban communities, rural communities have greater proportions of older people (aged 60+) with $\geq 2$ concurrent chronic medical conditions [4,5]. Furthermore, the transition from hospital to home constitutes a marked shift from provider-driven to self-managed care for rural patients. After hospital discharge, many patients and their families find themselves managing care previously provided by nurses in hospital (e.g., watching for worsening health conditions), typically with minimal training and professional guidance [6–8].

In our prior research, older rural medical patients at risk for hospital readmission and their families indicated that their most pressing unmet TC need was knowing how to recognize and address the signs of worsening health conditions [6–8]. Evidence indicates that patients typically receive little preparation in this regard and require more education prior to discharge on whether a sign or symptom necessitates professional intervention [9]. Such evidence is

particularly problematic given that older medical patients are increasingly discharged home more quickly and more ill due to a policy-emphasis on early discharge [10]. Moreover, compared to other patient populations, older medical patients have the highest hospital readmission rates in rural Ontario (65%) [3] and thus have a critical need for TC that emphasizes self-management.

Most TC models that foreground self-management emphasize the importance of educating patients on how to identify and respond to the signs of deteriorating health [1,2,11–15]. Results of TC trials have indicated that a warning signs intervention, as part of multicomponent hospital-to-home TC programs, reduces emergency department visits [16,17], hospital readmissions [1,15,17–20], and improves patient knowledge and confidence in managing changes in their health conditions [1] (all $p < .05$, small to moderate effect sizes).

Informed by a synthesis of the empirical literature [1,15–21], we developed a comprehensive description of a warning signs intervention [22]. The goal of the intervention is for patients to know the indicators of worsening health conditions and how to respond [22]. The intervention begins in hospital with an appraisal of patients' health literacy, their knowledge of the warning signs related to their condition(s), and their capacity to detect and address those signs at home, as well as what they need to learn to address them appropriately [22]. The intervention is delivered both prior to and after hospital discharge. To promote patient understanding, the intervention deploys the teach-back method, as well as educational materials that use lay language, symbols, and pictograms [22]. In our previous work, patients and their family caregivers rated the warning signs intervention as highly acceptable in preparing them to detect and respond to worsening health conditions [7], indicating that they would be very receptive to receiving it [23].

The warning signs intervention has great potential for widespread adoption in practice because it represents rural patients' most pressing unmet TC need, and patients and families are open to receiving it [23]. However, the Medical Research Council [24], other scholars [25] and intervention implementation frameworks advise assessing whether all potential users perceive an intervention as acceptable to avoid problems associated with low acceptability such as limited uptake and improper implementation [26,27], which reduce intervention effectiveness [28]. Acceptability pertains to the desirability of an intervention to prospective users and signifies positive perceptions of an intervention [29,30]. In the context of the warning signs intervention, acceptability reflects the extent to which potential users: anticipate the intervention to be appropriate for medical patients in detecting and responding to the signs of worsening health conditions; view it as convenient to use in the practice setting; perceive it as effective, relevant, applicable, and useful in helping medical patients manage the signs of worsening health conditions with minimal risks; currently employ or plan to employ the intervention; and are confident in their ability to deliver it [31].

Understanding the perceived acceptability of an intervention to potential users is critical because interventions perceived as acceptable are more likely to be adopted than those that are not perceived as such [32]. However, the perspectives of other healthcare providers involved in providing TC on the warning signs intervention remain unknown. Examining their perspectives is important because, although the warning signs intervention is primarily delivered by nurses, its implementation is inherently interprofessional and often involves collaboration with other healthcare providers (HCPs) [18]. For example, if nurses and other HCPs perceive the intervention to have limited relevance, widespread adoption is unlikely [7]. Similarly, if nurses and other HCPs lack confidence in their ability to deliver the intervention, then further work can be done to help develop their knowledge and skill to competently deliver it.

Additionally, home follow-up in rural communities is often conducted by a single HCP, who may not be a nurse, and this HCP may be required to implement the intervention

independently. Consultations with knowledge-users (administrative and clinical decision-makers) have confirmed that, due to limited human health resources in rural communities [33], single HCP follow-up is a common scenario. Therefore, comparing the perspectives of nurses and other HCPs (e.g., physical and occupational therapists, physicians) potentially involved in the delivery of the intervention is necessary prior to implementing it. For instance, nurses may perceive the intervention as acceptable and deliver it as designed but if other HCPs perceive it as unacceptable, they are unlikely to deliver it or may adapt the intervention to the extent that it no longer aligns with the one provided in trials, potentially rendering it ineffective [34]. Convergent perspectives on an intervention's acceptability support its implementation, while divergent perspectives underscore areas that need to be rectified prior to implementation [7].

## Study objective

This study examined and compared nurses' and other HCPs' perspectives on the acceptability of an evidence-based warning signs intervention proposed for rural hospital-to-home transitional care.

## Methods

### Design

This cross-sectional study was part of a larger multi-method project [22]. To maximize future uptake of the intervention in practice, the project was designed and conducted by academic researchers in collaboration with knowledge users (clinical and administrative leaders) [22]. In this paper, only the research methods used to assess nurses' and other HCPs' perceptions of the interventions' acceptability are described.

### Setting and sample

The study was conducted in Ontario, Canada and included English-speaking and -reading nurses and other HCPs who worked at least 21 hours/week in a hospital and/or community care (e.g., primary care) setting and self-identified as providing TC to rural patients with medical conditions. Recruitment, which was driven by the knowledge-users on our team in Southwestern and Northeastern Ontario by introducing the study at staff meetings, posting flyers, and distributing them through email, occurred from September 1st 2019 to May 1st, 2022. Recruitment took longer than anticipated because of the Covid-19 pandemic and had to be suspended several times at the sites where our knowledge-users were located. Consequently, we expanded recruitment to include all rural Ontario.

Participants who had initiated the survey, but did not complete it, were sent two reminders. Eligible participants were provided a $25 gift certificate for completing the survey. A minimum sample size of 21/group was needed to detect a large effect size (quantifying the differences in the mean acceptability ratings) between the two groups (nurses and other HCPs), setting the power at .80 and p-level at .05 [35].

### Data collection

Nurses and other HCPs read the intervention description and completed the study measures on Survey Monkey. The intervention acceptability attributes were measured by the Intervention Acceptability Scale [36] and the Treatment Acceptability Scale [31].

The Intervention Acceptability Scale, which has demonstrated internal consistency reliability (alpha > .80) [7,31] and factorial validity [31], contains seven self-report items reflected in

four subscales measuring participants' perspectives on the following intervention acceptability attributes: appropriateness (two items), effectiveness (two items), risks (one item), and ease of use (two items) with older rural patients in managing the signs of worsening health conditions. In the study sample, the scale had a Cronbach alpha of .82.

The other attributes of acceptability were measured by five self-report items from the Treatment Acceptability Scale, which was designed specifically for use with HCPs and has demonstrated content validity (indices were > 95% for several evidence-based interventions) and internal consistency reliability (moderate intercorrelations ranging from .4 to .6 among the items comprising the scale) [31]. These five items assessed participants' perceptions of the relevance and applicability of the intervention in helping to address the signs of worsening health conditions in the older rural patient population, confidence in their knowledge and skill to deliver it, and frequency of current use and likelihood of future use of the intervention in practice. In consultation with the scale developer, one item assessing "comfort" was revised to "confidence" in providing the intervention [32,37] and a fifth item assessing the perceived usefulness of the intervention was added. In the study sample, the scale had a Cronbach alpha of .81.

A five-point Likert-type scale that ranged from *not at all* (0) to *very much* (4) was used in the assessment of all items in both scales. The intervention was deemed acceptable if the attributes had a mean rating $\geq 2$, which is the response scale midpoint [31].

Demographic (e.g., age, gender) and professional (e.g., years of experience, professional designation, highest level of education) characteristics were measured using standard questions.

## Data analysis

Descriptive statistics, in accordance with each variable's level of measurement, were used to describe participants' demographic and professional characteristics. SPSS was used in the analysis. The mean and standard deviation were calculated for each subscale of the Intervention Acceptability Scale as well as each item of the Treatment Acceptability Scale.

Independent samples t-tests were used to compare nurses' and other HCPs' perceptions of the intervention on all attributes of intervention acceptability. Effect sizes (ES) were calculated as the standardized difference in the means of nurses' and other HCPs' scores on the acceptability attributes using Cohen's formula, with the ES interpretation of $< .20$ (no difference); .20 –.49 (small); .50–.79 (medium); and $> .80$ (large) [38].

## Ethical approval

Ethics approval was obtained from the Research Ethics Board at York University (certificate # e219-241) and from the Research Ethics Office at Health Sciences North (Project# 19–020). Written informed consent was obtained from all participants on an online platform prior to their completing the survey. All methods were carried out in accordance with the relevant guidelines and regulations of the World Health Organization's Declaration of Helsinki.

## Results

### Participant demographic and professional characteristics

A total of 110 HCPs initiated the survey but 28 did not meet the eligibility criteria. Of the 82 eligible HCPs, 5 withdrew implicitly by not answering any of the acceptability items or responding to follow-up requests to complete the survey. In total, 76 eligible HCPs completed the survey for a response rate of 93%, with a total sample size of 45 nurses (29 registered

nurses, 10 nurse practitioners, and 6 registered practical nurses), and 32 other HCPs (7 physicians, 6 social workers, 5 occupational and 5 physical therapists, 2 registered dieticians, 2 speech language pathologists, and 5 other).

As shown in Table 1, most nurses were women (95.6%) with an average age of 38.9 years (± 11.96). All self-identified as white. Most had a Baccalaureate degree as their highest level of education (44.4%), were working full-time hours (75.6%) in a hospital (53.3%) and had been practicing in their profession for an average of 10 years (range 1–50).

Other HCPs were also mostly women (78.1%) with an average age of 43.4 years (± 11.5). The majority self-identified as white (87.5%). Most had a Master's degree as their highest level of education (43.8%), were working in a hospital (68.8%), and had been practicing in their profession for an average of 13 years (range 1–38). All reported working full-time hours.

### Participant perceptions of intervention acceptability attributes

Table 2 presents the mean scores on the intervention acceptability attributes, which indicate, in general, participants viewed the intervention as acceptable. The mean scores on the Intervention Acceptability subscales and the items of the Treatment Acceptability Scale were greater than 2. Notable exceptions were convenience and frequency of current use of the intervention; the total sample's mean scores were less than 2 (the scale midpoint) on these attributes.

### Comparison of nurses' and other HCPs' perceptions of intervention acceptability attributes

As indicated in Table 2, there were no significant between-group (nurses and other HCPs) differences on perceived effectiveness, appropriateness, risks, convenience, usefulness, or relevance of the intervention, or in confidence in knowledge and skill in providing the intervention. However, statistically significant between-group mean differences were found for some of the other attributes of acceptability. Nurses' mean scores were higher than other HCPs on the perceived applicability of the intervention in helping patients detect and respond to the signs of worsening health conditions ($[t (75) = 2.95, p = .002, \eta^2 = .68]$), frequency of current use ($[t (75) = 2.52, p = .007, \eta^2 = .58]$), and likelihood of future use ($[t (75) = 2.95, p = .002, \eta^2 = .68]$) of the intervention. The differences were medium in magnitude.

## Discussion

Our study responds to growing calls to examine the perspectives of HCPs on the acceptability of proposed healthcare interventions before implementing them [26,27]. In general, participants ratings of all intervention acceptability attributes indicated positive perceptions. Consequently, the warning signs intervention represents a viable option for rural transitional care. However, exceptions were found for the attributes of convenience and frequency of current use of the intervention. The low ratings for these two attributes in the total sample may reflect concerns about increased workload and the limited resources available in rural areas. Given that early hospital discharge has become a priority in many jurisdictions [39], HCPs are increasingly likely to encounter high patient volume and turnover, and increased work tied to patients' admissions and discharges [39].

Prior research has highlighted that HCPs are frequently unable to prioritize transitional care due to time and resource constraints, and high patient acuity [40]. Hospital-based HCPs, for instance, may find it challenging to coordinate care with patients, informal caregivers, and interprofessional team members, some of whom practice in different locations (e.g., hospital, community) [40]. Transitional care necessitates communicating with these various team members, which may be difficult to do in busy practice environments [40]. Nonetheless, it is

**Table 1. Description of nurse and other healthcare provider demographic and professional characteristics.**

| Characteristics | Nurses (n = 45) | | Other healthcare providers (n = 32) | |
|---|---|---|---|---|
| | M | SD | M | SD |
| Age | 38.91 | 11.96 | 43.38 | 11.46 |
| | Md | R | Md | R |
| Years in Professional Practice | 10 | 1–50 | 13 | 1–38 |
| Year in Current Position | 6 | 1–50 | 8.5 | 1–33 |
| | n | Valid % | n | Valid % |
| Gender | | | | |
| Woman | 43 | 95.6 | 25 | 78.1 |
| Man | 2 | 4.4 | 7 | 21.9 |
| Ethnicity[¥] | | | | |
| White | 45 | 100 | 28 | 87.5 |
| Aboriginal | 1 | 2.2 | 0 | 0 |
| South Asian | | | 1 | 3.3 |
| Arab | | | 2 | 6.3 |
| Southeast Asian | | | 1 | 3.1 |
| Highest level of education | | | | |
| College diploma | 13 | 28.9 | 0 | 0 |
| Bachelor's degree | 20 | 44.4 | 10 | 31.3 |
| Medical degree | | | 5 | 15.6 |
| Other post-graduate certificate | | | 1 | 3.1 |
| Master's degree | 12 | 26.7 | 14 | 43.8 |
| PhD | | | 2 | 6.3 |
| Hours worked | | | | |
| Full time hours | 34 | 75.6 | 32 | 100 |
| Part time hours | 11 | 24.4 | 0 | 0 |
| Professional designation | | | | |
| Registered Practical Nurse | 6 | 13.4 | | |
| Registered Nurse | 29 | 64.4 | | |
| Nurse practitioner | 10 | 22.2 | | |
| Physician | | | 7 | 21.9 |
| Social Worker | | | 6 | 18.8 |
| Occupational Therapist | | | 5 | 15.6 |
| Physical Therapist | | | 5 | 15.6 |
| Speech Language Pathologist | | | 2 | 6.3 |
| Other | | | 5 | 15.6 |
| Healthcare sector[¥] | | | | |
| Hospital | 24 | 53.3 | 22 | 68.8 |
| Community | 21 | 46.7 | 17 | 53.1 |
| Primary position[¥] | | | | |
| Staff Nurse | 22 | 48.9 | | |
| NP | 7 | 15.6 | | |
| Rapid Response Nurse | 4 | 8.9 | | |
| Care Coordinator | 3 | 6.7 | 3 | 9.4 |
| Primary Care NP | 2 | 4.4 | | |
| Team Leader | 2 | 4.4 | | |
| Occupational Therapist | | | 5 | 15.6 |
| Physician | | | 4 | 12.5 |

*(Continued)*

**Table 1.** (Continued)

| Characteristics | Nurses (n = 45) | | Other healthcare providers (n = 32) | |
|---|---|---|---|---|
| Physiotherapist | | | 4 | 12.5 |
| Social Worker | | | 3 | 9.4 |
| Registered Dietician | | | 2 | 6.6 |
| Speech Language Pathologist | | | 2 | 6.6 |
| Other | 5 | 11.0 | 9 | 27.9 |

*Notes.* [¥]Total for these variables is $\neq$ 45 for nurses or 32 for other healthcare providers because a respondent could select more than one option. M = mean.

SD = standard deviation. NP = Nurse Practitioner. PhD = Doctor of Philosophy.

troubling that the results indicated low frequency of current use of the intervention. Since rural dwellers have high expectations for their preparedness to detect and respond to signs of worsening health conditions because of their limited access to HCPs and travel distances to hospitals [6,8], a greater emphasis on such preparation appears to be much needed.

In terms of between-group differences on three attributes of intervention acceptability (nurses' ratings were consistently higher than other HCPs on applicability, frequency of current use, and the likelihood of future use of the intervention), we consider several possible contextual explanations. It is possible that, although education on warning signs for patients appears to be limited, what education is provided is likely delivered predominantly by nurses and consequently nurses are likely more familiar with the intervention than are other HCPs. Knowledge of, or experience with, an intervention has been associated with more favorable opinions of an intervention's applicability [41]. In a similar vein, it is also possible that nurses' prior experience with this type of care may explain their higher ratings on frequency of current

**Table 2. Acceptability attributes ratings on the warning signs intervention for total sample, nurses, and other healthcare providers.**

| Scale | Acceptability attributes ratings[†] | | | | | | | |
|---|---|---|---|---|---|---|---|---|
| | Total sample (n = 77) M (SD) | % with score $\geq$ 2 | Nurses (n = 45) M (SD) | % with score $\geq$ 2 | Other healthcare providers (n = 32) M (SD) | % with score $\geq$ 2 | t-test | Cohen's d (difference in group means) |
| Intervention acceptability subscales | | | | | | | | |
| Effectiveness | 2.35 (.85) | 77% | 2.43 (.82) | 80% | 2.23 (.88) | 72% | 1.02 | .24 |
| Appropriateness | 2.43 (.72) | 87% | 2.51 (.63) | 91.1% | 2.31 (.83) | 81% | 1.20 | .28 |
| Risks | 3.71 (.60) | 99% | 3.71 (.55) | 100% | 3.72 (.68) | 97% | -.05 | -.01 |
| Convenience | 1.78 (.73) | 49% | 1.82 (.70) | 51.1% | 1.75 (.78) | 47% | .29 | .07 |
| Treatment Acceptability Scale items | | | | | | | | |
| Usefulness | 2.42 (.95) | 81% | 2.51 (.84) | 86.7% | 2.28 (1.09) | 72% | 1.05 | .24 |
| Confident in knowledge and skill | 2.23 (1.06) | 77% | 2.38 (.89) | 82.2% | 2.03 (1.26) | 69% | 1.42 | .33 |
| Relevance | 2.48 (.81) | 87% | 2.60 (.65) | 95.6% | 2.31 (.97) | 75% | [¥]1.46 | .36 |
| Applicability | 2.39 (.86) | 86% | 2.62 (.75) | 93.3% | 2.06 (.91) | 75% | 2.95 | .68 |
| Frequency of current use | 1.53 (1.14) | 58% | 1.80 (1.14) | 64.4% | 1.16 (1.05) | 50% | 2.52 | .58 |
| Likelihood of future use | 2.01 (.94) | 68% | 2.27 (.81) | 80% | 1.66 (1.00) | 50% | 2.95 | .68 |

Notes.

[†]Equal variance not assumed. M = mean. SD = standard deviation.

use of the intervention and likelihood of using it in the future. There is some evidence to support that HCPs who are knowledgeable about an intervention are more likely to use the intervention in practice [42].

Since nurses tend to be the main professional responsible for patient education about signs of worsening health conditions, it is possible that the nurses in our study may have rated the applicability of the warning signs intervention higher than other HCPs because they saw the intervention's potential to improve the lives of patients. Other research has identified a similar cleavage in the views of nurses and other HCPs. For instance, in comparing nurses' and other HCPs' responses on the clinical value and utility of four different glucose monitoring devices, Greenwood and Grady found that nurses consistently differed in their ratings of the extent to which the devices could assist patients in managing diabetes; the authors concluded that the differences in responses could be explained by the fact that nurses tend to help patients select a device and train patients on it use, and other HCPs do not [43]. Likewise, the nurses in our study sample may have viewed the applicability of the warning signs intervention to be greater, compared to other HCPs, because nurses tend to be the ones who provide related patient education.

It is possible that other HCPs' perception of their role may have contributed to their lower ratings in applicability, frequency of current use, and likelihood of future use of the intervention. There is some research indicating that other HCPs tend to view their role as specific and circumscribed whereas nurses perceive their role as more holistic [44]. The reluctance of other HCPs to deliver the intervention may be rooted in their theoretical orientation and clinical experience, as some literature has shown that HCPs deliver interventions they view favorably (i.e., are acceptable, advantageous, and consistent with their theoretical orientation and with their clinical experiences) [45,46]. It is conceivable that, based on their training and socialization into their professional role, other HCPs were not comfortable with the possibility of assuming tasks that they perceive to be outside their scope of practice. In other words, HCPs' lower rating on the likelihood of using the intervention in the future, compared to nurses' higher rating, may be indicative of HCPs being less motivated to assume tasks they see as outside of their professional role and responsibilities. Prior research has identified that redeployment of HCPs to areas of work outside their scope of practice or beyond their usual activities is a source of stress [47], so it is reasonable to assume that many other HCPs would avoid doing so.

Similarly, different groups may have different motivations when considering an intervention, which could influence their ratings of it [48]. Nurses may have been focusing on the interventions' capacity to validate their roles as patient educators and spending more time in related activities, whereas, as discussed above, other HCPs may have been wary of the intervention's potential to increase their workload.

## Implications for future research

Given that both nurses and other HCPs perceived the warning signs intervention as acceptable supports incorporating it into rural transitional care and, if there are concerns about its effectiveness in rural communities where testing has been limited, conducting effectiveness studies [49]. Further research is warranted to validate the reasons we proposed for the total sample's low ratings on convenience and frequency of current use of the intervention as well as the between-group differences in ratings on the applicability, frequency of current use, and likelihood of future use of the intervention. Researchers may explore features of the intervention that may account for low ratings in these two attributes and how these features may be addressed. Such research is imperative given that many patients do not know the signs of

worsening health conditions after hospital discharge [9,50] which is particularly concerning in rural communities where patients tend to travel far distances to access limited human health resources and deaths from treatable health conditions are significantly higher in more remote areas [51].

## Implications for practice and policy

The impetus for this study was healthcare administrators' recognition of the need to have other HCPs deliver the warning signs intervention given the limited human health resources in rural communities. While we acknowledge that all categories of professionals in our other HCP group may not be able to provide the warning signs intervention on their own due to ethical and regulatory considerations, we nonetheless stress that the warning signs intervention is necessarily interprofessional. At minimum, other HCPs could support the implementation of the intervention, even if they cannot be responsible for its overall delivery. While limited human health resources are often the rationale for having HCPs take on new roles, workforce shortages are key barriers to taking on new roles such as delivering the warning signs intervention [52]. Consequently, administrators need to ensure sufficient staffing when introducing new roles and allow adequate time for HCPs to master them [52]. As well, administrators would need to provide training and support in how to deliver the warning signs intervention which has been provided successfully in specialized geriatrics teams. For example, interprofessional team members have been trained to perform geriatric assessments despite this role not having been traditionally assigned to them [53]. Such cross-disciplinary training could be used in the implementation of the warning signs intervention and could mitigate tensions arising from professional role perceptions.

## Study limitations

This study was conducted in one Canadian province. The findings may not be generalizable outside of Ontario, Canada, but other researchers may use our systematic approach to compare nurses' and other HCPs' perspectives on the acceptability of the warning signs interventions in their jurisdiction.

## Conclusions

Overall, nurses and other HCPs perceived the warning signs intervention positively. The intervention thus represents a tenable option for rural transitional care in Ontario, Canada, and possibly other jurisdictions emphasizing TC. We identified medium between-group differences in perceived acceptability in three attributes, which we propose are accounted for by contextual factors. More research is needed to validate these propositions and further explore HCPs' perspectives.

## Author Contributions

**Conceptualization:** Mary T. Fox.

**Formal analysis:** Mary T. Fox.

**Funding acquisition:** Mary T. Fox.

**Investigation:** Mary T. Fox.

**Methodology:** Mary T. Fox.

**Project administration:** Jeffrey I. Butler.

**Validation:** Mary T. Fox.

**Writing – original draft:** Mary T. Fox.

**Writing – review & editing:** Mary T. Fox, Jeffrey I. Butler, Adam M. B. Day, Evelyne Durocher, Behdin Nowrouzi-Kia, Souraya Sidani, Ilo-Katryn Maimets, Sherry Dahlke, Janet Yamada.

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
