## [Decision Letter · Decision Letter 0]

8 Feb 2024

Healthcare providers’ perceived acceptability of a warning signs intervention for rural hospital-to-home transitional care: A cross-sectional study

PONE-D-23-30096

Dear Dr. Fox,

We’re pleased to inform you that your manuscript has been judged scientifically suitable for publication and will be formally accepted for publication once it meets all outstanding technical requirements.

Kind regards,

Stefan Grosek, Ph.D., M.D.,

Academic Editor

PLOS ONE

dditional Editor Comments (optional):

Reviewers' comments:

Reviewer's Responses to Questions

**Comments to the Author**

1. Is the manuscript technically sound, and do the data support the conclusions?

Reviewer #1: Yes

Reviewer #2: Yes

2. Has the statistical analysis been performed appropriately and rigorously? 

Reviewer #1: Yes

Reviewer #2: Yes

3. Have the authors made all data underlying the findings in their manuscript fully available?

Reviewer #1: Yes

Reviewer #2: No

4. Is the manuscript presented in an intelligible fashion and written in standard English?

Reviewer #1: Yes

Reviewer #2: Yes

5. Review Comments to the Author

Reviewer #1: Thank you for the opportunity to review PONE-D-23-30096 Healthcare providers’ perceived acceptability of a warning signs intervention for rural hospital-to-home transitional care: A cross-sectional study. The authors’ stated purpose was to examine and compare nurses and other healthcare providers’ perceived acceptability of an evidence-based warning signs intervention proposed for rural transitional care. This is a prolific group of researchers who have written extensively on the subject. The researchers administered a survey to health care workers and analyzed the results, reflecting on the significance of the answers. This manuscript appears to be the related to the protocol described in the author’s previous article entitled ‘Collaborating with healthcare providers to understand their perspectives on a hospital-to-home warning signs intervention for rural transitional care: protocol of a multimethod descriptive study’, although it appears the focus groups part of the proposed protocol was not completed or is not included here. 76 eligible HCPs completed the survey for a response rate of 93%, with a total sample size of 45 nurses (29 registered nurses, 10 nurse practitioners, and 6 registered practical nurses), and 32 other HCPs (7 physicians, 6 social workers, 5 occupational and 5 physical therapists, 2 registered dieticians, 2 speech language pathologists, and 5 other).

The main claims of the paper are that in general, participants viewed the intervention as acceptable, except for convenience and frequency of use of the intervention.

The data and analyses generally support the claims made by the authors. The manuscript is readable and concise.

The authors divided their subjects into two groups, one being nurses and the comparison being a heterogeneous group of physicians, occupational therapists, physical therapists, dieticians, speech language pathologists, and 5 ‘other. With such wide-ranging roles and educational backgrounds and skillsets, it is not a group with internal similarity. From the original protocol, the plan was for 3 groups; nurses, physicians, and allied health professionals, but I assume there were not enough numbers in each group to power a 3-way analysis.

It is unfortunate that the number of subjects was too small to divide into 3 groups. One would certainly expect that different healthcare workers would view a teaching task such as this differently, given their respective disciplines and roles. The paper could be used as data to support the need for more nurses to perform patient visits in rural areas.

The authors acknowledged that the study is very limited in its applicability to other settings, especially given the small sample size.

Because it is small and compares one group with another heterogeneous group, it does not contribute very much to our overall understanding of the problem, but builds on the previous work by the authors and will undoubtedly lead to more study.

Reviewer #2: The manuscript is on a very relevant topic: improving transitional care from hospital to home. While a relevant intervention has been well described previously, this study adds an examination of the tolerability of the intervention among health care providers. The topic is well introduced and comprehensible. The current situation and its gaps were well explained and the aim of the research is well motivated. Less clear was the aspect of rurality. While it was well motivated why rural communities are vulnerable and need to be specifically addressed, it is less clear how this translates into the proposed intervention: how does the intervention address the aspect of rurality or was it just an evaluation of a generic intervention in a rural setting? I find the aggregation of the very different HCPs into one group compared to nurses rather problematic because of their heterogeneity. I also felt that this broad aggregation was unnecessary.

In summary, this is well-conducted, well-communicated research. It adds to the literature on a relevant topic. There are only a few minor comments for improvement (in addition to those mentioned above):

- Lines 161ff. Sample size determination: What difference in mean acceptability ratings was anticipated? Citation nr. 35 is unnecessary.

- Line 214: What are the five other health care professions?

- Line 241 "there were no significant between-group": I assume the word "difference" is missing.

- Line 332 Bad wording: human health human resources

6. PLOS authors have the option to publish the peer review history of their article (what does this mean?). If published, this will include your full peer review and any attached files.

Reviewer #1: No

Reviewer #2: No

---

## [Editor Report · Acceptance letter]

21 Feb 2024

PONE-D-23-30096 

PLOS ONE

Dear Dr. Fox, 

I'm pleased to inform you that your manuscript has been deemed suitable for publication in PLOS ONE. Congratulations! Your manuscript is now being handed over to our production team.

Kind regards, 

on behalf of

Professor Stefan Grosek 

Academic Editor

PLOS ONE